# Immunoprofiles and Oncologic Outcomes of 15 Patients with Androgen Receptor-Positive Salivary Duct Carcinoma

**DOI:** 10.3390/cancers16061204

**Published:** 2024-03-19

**Authors:** Emile Gogineni, Blake E. Sells, Khaled Dibs, Sachin R. Jhawar, Catherine T. Haring, Abberly L. Limbach, David J. Konieczkowski, Sung J. Ma, Simeng Zhu, Sujith Baliga, Darrion L. Mitchell, John C. Grecula, Marcelo Bonomi, Priyanka Bhateja, Matthew O. Old, Nolan B. Seim, Stephen Y. Kang, James W. Rocco, Arnab Chakravarti, Dukagjin M. Blakaj, Mauricio E. Gamez

**Affiliations:** 1Department of Radiation Oncology, The Ohio State University Wexner Medical Center, 460 W. 10th Ave, Columbus, OH 43210, USA; khaled.dibs2@osumc.edu (K.D.); sachin.jhawar@osumc.edu (S.R.J.); david.konieczkowski@osumc.edu (D.J.K.); sungjun.ma@osumc.edu (S.J.M.); simeng.zhu@osumc.edu (S.Z.); sujith.baliga@osumc.edu (S.B.); darrion.mitchell@osumc.edu (D.L.M.); john.grecula@osumc.edu (J.C.G.); arnab.chakravarti@osumc.edu (A.C.); dukagjin.blakaj@osumc.edu (D.M.B.); 2St. Louis School of Medicine, St. Louis, MO 63310, USA; blake.sells@wustl.edu; 3Department of Otolaryngology, The Ohio State University Wexner Medical Center, 460 W. 10th Ave, Columbus, OH 43210, USA; catherine.haring@osumc.edu (C.T.H.); matthew.old@osumc.edu (M.O.O.); nolan.seim@osumc.edu (N.B.S.); stephen.kang@osumc.edu (S.Y.K.); james.rocco@osumc.edu (J.W.R.); 4Department of Pathology, The Ohio State University Wexner Medical Center, 460 W. 10th Ave, Columbus, OH 43210, USA; abberly.lottlimbach@osumc.edu; 5Department of Medical Oncology, The Ohio State University Wexner Medical Center, 460 W. 10th Ave, Columbus, OH 43210, USA; marcelo.bonomi@osumc.edu (M.B.); priyanka.bhateja@osumc.edu (P.B.); 6Department of Radiation Oncology, Mayo Clinic, 200 First St SW, Rochester, MN 55905, USA; gamezharo.mauricio@mayo.edu

**Keywords:** salivary duct carcinoma (SDC), radiation, intensity-modulated radiation therapy (IMRT), androgen receptor (AR), HER-2/Neu, ERRB2, PI3KCA, estrogen receptor (ER), head and neck cancer (HNC), molecular, genomic, gene, profile, immunoprofiling, targeted therapy, systemic therapy, GnRH, androgen deprivation therapy (ADT), Herceptin, leuprolide, bicalutamide

## Abstract

**Simple Summary:**

Salivary duct carcinomas (SDC) can present with distinct immunologic profiles similar to breast cancer, such as androgen receptor (AR) and HER-2/Neu-positivity, raising the hypothesis that these tumors may respond to hormonal signaling. No consensus exists on how to best manage this entity. The data evaluating the use of targeted therapies, such as androgen deprivation therapy and HER-2 receptor inhibitors, in the front-line setting when treating curatively is limited. We studied patients with AR+ SDC, demonstrating high rates of control and survival using an aggressive approach to treatment. Immunoprofiles were highly variable, highlighting the potential for future treatment individualization. We hope that this may allow for personalization of treatment in the future, using molecular profiling to determine whether the addition of biological agents in the definitive setting against specific targets, such as AR and HER-2/Neu, will further improve outcomes for these patients.

**Abstract:**

Background: Salivary duct carcinomas (SDC) are a rare and aggressive subtype of salivary gland neoplasm. They can present with distinct immunoprofiles, such as androgen receptor (AR) and HER-2/Neu-positivity. To date, no consensus exists on how to best manage this entity. Methods: All patients diagnosed with nonmetastatic AR+ SDC of the parotid from 2013 to 2019 treated with curative intent were included. Immunologic tumor profiling was conducted using 24 distinct markers. Kaplan–Meier analyses were used to estimate locoregional recurrence (LRR), distant control, and overall survival (OS). Results: Fifteen patients were included. Nine (60%) patients presented with T4 disease and eight (53%) had positive ipsilateral cervical lymphadenopathy. Ten (67%) patients underwent trimodality therapy, including surgery followed by adjuvant radiation and concurrent systemic therapy. The median follow-up was 5.5 years (interquartile range, 4.8–6.1). The estimated 5-year rates of LRR, distant progression, and OS were 6%, 13%, and 87%, respectively. Conclusion: Despite only including AR+ SDC of the parotid, immunoprofiles, such as expression of HER-2, were highly variable, highlighting the potential to tailor systemic regimens based on individual histologic profiles in the future. Studies with larger patient numbers using tumor-specific molecular profiling and tumor heterogeneity analyses are justified to better understand the biology of these tumors. Molecularly informed treatment approaches, including the potential use of AR- and HER-2/Neu-directed therapies upfront in the definitive setting, may hold future promise to further improve outcomes for these patients.

## 1. Introduction

Salivary duct carcinoma (SDC) is a rare cancer that originates from the excretory ducts of the salivary glands [1]. It represents less than 5% of all diagnosed malignant salivary gland neoplasms [2,3]. The incidence of SDC is highest in older men, and the majority of cases originate in the parotid gland [4].

This tumor was first described in 1968 as a group of salivary gland tumors similar in histopathological appearance to ductal carcinoma of the breast, raising the hypothesis that these tumors may respond to hormonal signaling [5,6]. Recent studies investigating molecular and histologic profiles of SDC report that the androgen receptor (AR) signaling pathway is involved in tumor progression and aggressiveness [7]. The immunoprofiles of these tumors suggests that clinical outcome and response to treatment may depend on the receptor status of AR, HER-2/Neu, and PIK3CA, among others [8]. Other studies suggest that apocrine phenotype and AR expression are requisites for diagnosis of SDC, proposing that salivary malignancies that do not fit these criteria may be reclassified as alternative cancer subtypes [9]. Correspondingly, routine testing of AR and HER-2/Neu in patients with SDC at the time of diagnosis may allow for prognostic and therapeutic stratification, as the presence of these markers may serve as therapeutic targets, providing a potential path towards improving outcomes [10].

Previous retrospective studies have demonstrated poor progression-free and overall survival (OS) for patients with SDC, with five-year survival ranging from 42% to 55% [8,11,12,13]. Unfortunately, the rarity of this tumor limits our ability to understand responsible molecular drivers and develop effective treatment strategies. Currently, treatment typically consists of surgery with or without adjuvant radiation alone or in combination with chemotherapy, regardless of immunoprofile. The National Comprehensive Cancer Network (NCCN) Guidelines Version 2.2024 discuss consideration for molecular profiling in patients with distant metastases but do not individualize treatment based on histologic features, tumor mutational burden, molecular profile, or biomarker status in the upfront setting when treating SDC or any other salivary gland tumor curatively [14].

This highlights the need for ongoing study of patients with SDC to further understand immunoprofiles and differences in clinical outcomes based on receptor status. In this analysis, we aimed to assess and report clinical outcomes, patterns of failure to treatment, and immunoprofiles for patients diagnosed with AR+ SDC of the parotid gland treated with curative intent. We report results for histologic/immunohistochemistry testing, locoregional recurrence (LRR), distant progression, OS, and toxicity in our cohort of patients.

## 2. Materials and Methods

The study was conducted in accordance with the World Medical Association’s Declaration of Helsinki and approved by our Institutional Review Board. All patients diagnosed with AR+ SDC of the parotid between 2013 and 2019 at our National Cancer Institute-Designated Cancer Center treated with curative intent were included. Patients with metastatic disease at presentation, those who did not complete their full course of radiation, and those with alternative histology or negative AR status were excluded.

All diagnoses were confirmed histopathologically after evaluation by an expert head and neck pathologist at our institution. All patients were initially staged with computerized tomography (CT) and/or magnetic resonance imaging (MRI) of head and neck, in addition to CT chest and/or positron emission tomography/computed tomography (PET/CT). Staging was conducted in accordance with the American Joint Committee on Cancer (AJCC) Cancer Staging Manual, Seventh Edition, as was standard at the time. Staging was based on pathologic findings for those who underwent surgery, while clinical staging was used for those who did not. Treatment recommendations were determined by an institutional multidisciplinary head and neck tumor board based on surgical candidacy, stage, and presence of high-risk pathologic features for those who underwent surgery, including surgical margins, lymphovascular (LVI) and perineural (PNI) invasion, and the presence/number of involved lymph nodes with or without extranodal extension. All patients eligible for surgery underwent surgical resection. The extent of surgical resection was guided by the disease burden but included radical parotidectomy with ipsilateral neck dissection ± resection of adjacent soft issues, including the overlying skin and/or facial nerve. The use of adjuvant radiation ± systemic therapy was determined by a multidisciplinary tumor board discussion based on clinical and pathologic features. HER-2/Neu staining was considered positive with an immunohistochemistry (IHC) score of 3+. The absence of a particular marker was registered in our database as missing information.

For radiation planning purposes, all patients underwent CT simulation in the supine position with a head and neck immobilization device using thermoplastic mask with or without a custom-made mouthpiece for tongue deviation. All surgical scars were wired. The use of bolus was determined by the treating radiation oncologist and was typically employed for patients with skin invasion. Relevant preoperative imaging was fused to CT simulation scans at the discretion of the treating physician.

Clinical target volumes (CTV) were determined using preoperative staging scans, operative notes, final pathology findings, and discussion with the head and neck surgeon. The use of elective ipsilateral nodal irradiation was determined by the treating physician based on tumor stage, presence of high-risk pathologic features, and number of lymph nodes dissected. CTVs included ipsilateral levels Ib-IV for N0 patients treated with elective nodal irradiation, while CTVs included levels Ib-V for N+ patients. Planning target volumes (PTV) were created by adding a 3–5 mm isometric expansion from CTV to account for setup uncertainties, cropped 3 mm from skin for those without skin invasion. All patients were treated with intensity-modulated radiation therapy (IMRT) using volumetric arc therapy (VMAT) techniques. Prescribed radiation doses included 6000–6996 cGy to the primary tumor or surgical bed, 5940–6000 cGy to involved lymph node stations ± one station above and below those involved, and 5400–5600 cGy to uninvolved elective nodal stations. In all cases, radiation treatment plans were evaluated and approved during weekly head and neck chart rounds review. Image-guided radiation therapy (IGRT) was employed for all patients using daily ConeBeam CT (CBCT) ± orthogonal films.

Upon treatment completion, patients were routinely followed every 3 months for the first year, every 4 months for the second year, and every 6 months thereafter. At each follow-up visit, a detailed history and physical examination was taken. All toxicities were reported per the Common Terminology Criteria for Adverse Events (CTCAE) version 4.0 if present. Late toxicities were defined as toxicities present at least three months after the conclusion of treatment. Restaging imaging studies were performed at 3 months after treatment completion and when clinically indicated thereafter.

### Statistical Analysis

Summary statistics for patient characteristics are presented as median and interquartile ranges (IQR). Kaplan–Meier curves were generated to display LRR, distant progression, and OS curves, using the time from the date of diagnosis to the date of event. No comparisons of subgroups were performed due to small sample size. Data analysis was performed using SPSS statistical software version 25.0 (IBM Corp, Armonk, NY, USA).

## 3. Results

A total of 16 patients were identified with AR+ SDC of the parotid gland. One patient was excluded due to completing only 1600 of his planned 6000 cGy of radiation due to unrelated medical complications, after which he was lost to follow-up. Fifteen patients were included in the final analysis. Patient demographic characteristics and treatment regimens are included in Table 1.

The median age was 62 years (IQR, 57–77). Nine (60%) patients presented with T4 disease, and eight (53%) presented with positive ipsilateral cervical lymphadenopathy. Along with the evaluated clinical features, immunoprofiling of all patients was conducted. Examples of histologic pathology and stains are shown in Figure 1.

All patients were AR-positive. Additional immunostains included AE1/3 (67%), CK7 (60%), GATA3 (47%), mammaglobin (33%), ER (27%), and HER-2/Neu (20%). To better visualize the variability in immunoprofiles, immunohistochemistry results from each tumor are provided in Figure 2.

Fourteen (93%) patients underwent radical resection of the primary tumor with ipsilateral selective lymph node dissection followed by adjuvant radiation, while one (7%) patient received definitive radiation with concurrent androgen-deprivation therapy (ADT) due to surgical ineligibility from disease extension (T4b disease) and extensive comorbidities. Of the fourteen patients who underwent surgery, eight (57%) had positive margins, four (29%) had margins < 1 mm, and five (36%) had extranodal extension. Radiation was delivered to the primary site in all cases and to the ipsilateral neck for twelve (80%) patients (three patients had minimal risk factors for nodal recurrence and a large number of dissected nodes that were negative for pathologic involvement). Ten (67%) patients received systemic therapy concurrent with radiation, including Cisplatin (*n* = 3, 20%), Carboplatin + Paclitaxel (*n* = 3, 20%), Carboplatin (*n* = 2, 13%), and gonadotropin releasing hormone (GnRH) analog (*n* = 2, 13%), while one (7%) patient received salvage non-steroidal androgen receptor inhibitor after progression.

Median follow-up was 5.5 years (IQR, 4.8–6.1). The estimated 5-year rates of LRR, distant progression, and OS were 6%, 13%, and 87%, respectively, as shown in Figure 3.

Two patients experienced recurrence at a median of 0.8 years (range, 0.4–1.1), including one with local and distant progression and one with distant progression only. No patient experienced LRR without distant progression. A 56-year-old male with hepatitis C and liver cirrhosis presented with pT4aN0 SDC of the right parotid and underwent surgical resection and selective lymph node dissection. Pathology showed a <1 mm deep margin and +LVI. AR, mammaglobin, AE1/3, and MNF-116 were positive. He completed a course of adjuvant radiation alone to 6600 cGy in 33 fractions. He had documented local and distant failure (bone metastasis) 0.4 years after diagnosis and was started on a salvage nonsteroidal androgen receptor inhibitor but died from further progression shortly thereafter. A 78-year-old female patient had a history of ductal carcinoma in situ of the right breast with an unknown molecular profile treated with lumpectomy. She was subsequently diagnosed with pT4aN2b SDC of the right parotid with skin involvement. She underwent primary surgical resection and lymph node dissection. Pathology showed positive microscopic margins, +PNI, +LVI, and 34/34 positive lymph nodes. AR, BRST-2, CK7, EMA, and AE1/3 were positive. She received adjuvant radiation to a dose of 6600 cGy in 33 fractions with concurrent weekly Cisplatin. She experienced distant progression with bone and lung metastasis without LRR 1.1 years after diagnosis and died shortly thereafter.

Table 2 captures acute and late toxicities. The most frequent acute grade ≥ 2 toxicities included pain (*n* = 11, 73%), mucositis (*n* = 11, 73%), dysgeusia (*n* = 11, 73%), radiation dermatitis (*n* = 9, 60%), and xerostomia (*n* = 8, 53%). The most common late grade ≥ 2 toxicity was pain (*n* = 3, 20%). Only one (7%) patient required PEG tube placement, an 81-year-old male with T4bN2b disease who presented with facial paralysis and trismus treated with definitive radiation to 6996 cGy in 33 fractions with concurrent and adjuvant GnRH analog. He remains PEG tube-dependent on last follow-up evaluation. No grade 4 or 5 toxicities were observed.

## 4. Discussion

We reviewed our institutional experience with AR-positive SDC of the parotid gland, assessing immunoprofiles, treatment characteristics, and clinical outcomes. Despite poor prognostic features, with ≥50% of patients having T4 and N+ stage disease, positive margins, +PNI, and +LVI, our study demonstrated high rates of control and survival using a multidisciplinary treatment approach. This included 67% of patients receiving trimodality treatment, including surgery followed by adjuvant radiation to ≥6600 cGy concurrent with systemic therapy. The 5-year rates of LRR, distant progression, and OS in our cohort were 6%, 13%, and 87%, respectively, while rates of toxicity were acceptably low.

Oncologic and toxicity outcomes from our series compared favorably to other studies reporting outcomes of SDC, as shown in Table 3 [8,13,15,16,17,18,19]. These favorable outcomes may be related to the homogeneity of patients included in our study and may partially be due to our approach of treatment escalation, with a large percentage of patients receiving trimodality therapy. As shown in Table 3, >50% of patients treated in our cohort received trimodality therapy, while only 20–30% of patients received this approach of treatment escalation in most other studies. We also treated with upfront ADT in two patients, while other studies reporting utilization of this systemic therapy reserved its use for patients who progressed after treatment. Despite this treatment escalation, rates of toxicity in our study were similar to those reported in the literature, which showed 20–30% acute grade 3 toxicities and 5–10% late grade 3 toxicities [19,20].

Despite only including AR+ SDC of the parotid, immunoprofiles, such as expression of HER-2, were highly variable. Much of the previously published clinical data is similarly retrospective in nature but frequently does not include in-depth analysis of immunologic tumor profiles and detailed descriptions of treatment, including radiation dose, volumes, and systemic therapy.

Like a large majority of data reporting outcomes for SDC, our study showed distant metastasis as the most common pattern of failure. This highlights both the importance of understanding genetic, molecular, and transcriptomic drivers of disease progression and resistance to therapy and the need to identify more effective systemic therapy strategies.

### 4.1. Androgen Receptor and Treatment with Androgen Deprivation Therapy

Recent studies have identified AR expression in more than 75% of patients with SDC [17]. Others, such as that by Williams et al., show AR expression in >97% of SDC on IHC, suggesting that all AR-negative tumors initially diagnosed as SDC may be reclassified upon further pathologic review [9]. They propose that apocrine phenotype and AR expression define SDC and that salivary malignancies which do not fit these criteria can almost always be reclassified as alternative cancer subtypes. Thus, oncologic outcomes from studies that include patients with AR-negative SDC should be interpreted with caution when evaluating prognosis and response to treatment for SDC.

AR is a nuclear hormone receptor that regulates the transcription of many effector genes involved in cell proliferation, invasion, and survival. It is also identified in the tumorigenesis of many cancers, such as breast and prostate cancer [7,21]. ADT has been extensively studied in other malignancies, such as prostate cancer [22]. In addition to direct tumor kill of AR+ cancer cells through apoptosis and extending doubling time by shifting actively dividing cells to quiescence, ADT plays a role in tumor vascularization and angiogenesis [23,24,25,26]. Tumor growth and metastasis depend on angiogenesis [27]. Poor vascularization can lead to hypoxia, which is associated with recurrence and poor prognosis [28,29]. These changes caused by ADT on the cellular level create a synergistic effect when combined with RT through radiosensitization. ADT has been shown to significantly reduce the dose of RT required to eradicate 50% of AR+ prostate cancer [30,31]. The timing and sequencing of ADT and radiation can also impact outcomes, as delivery of ADT neoadjuvant and concurrent with RT has been shown to have a greater effect than when delivered after RT [30,31].

Given SDC’s AR positivity and resistance to chemotherapy [17], ADT has emerged as a promising treatment option for SDC. A Dutch case series reported outcomes for 35 patients with distant metastases or incurable locally advanced AR+ SDC treated with ADT as first-line palliative therapy versus best supportive care [32]. After a median follow-up of 10 months, the median OS for ADT-treated patients was 17 months, and 29 months for those showing clinical benefit, versus 5 months for those only receiving best supportive care. Due to the observed clinical benefit, the authors concluded that ADT should be recommended in advanced AR+ SDC.

While ADT has been studied in the adjuvant setting and after progression, there are no studies to our knowledge evaluating its use concurrently with radiation as first-line therapy for SDC. Two (13%) patients in our cohort were treated with ADT concurrent with RT. One was an 81-year-old male with an extensive smoking history and cT4bN2b SDC of the right parotid. AR and AE1/3 were positive. He was not eligible for surgery or chemotherapy due to disease extent and comorbidities and underwent definitive radiation to 6996 cGy in 33 fractions concurrent with leuprolide. The other was a 58-year-old male with pT4bN3b SDC of the right parotid, who underwent surgical resection and selective lymph node dissection. Pathology showed positive margins, +PNI, +LVI, and extranodal extension. AR, CK7, AE1/3, and GATA3 were positive. He completed a course of adjuvant radiation to 6600 cGy in 33 fractions concurrent with leuprolide. Despite their extensive medical and oncologic risk factors, they remain alive without evidence of disease after 6.3 and 5.5 years of follow-up, respectively. We believe this treatment approach represents a promising area of investigation. Several ongoing clinical trials are assessing the response of locally advanced, unresectable, recurrent, and metastatic AR+ SDC patients to Enzalutamide, Bicalutamide + Triptorelin or Abiraterone (NCT02749903, NCT01969578, and NCT02867852).

### 4.2. Resemblance of SDC to Breast Cancer, Association with HER-2/Neu, and Treatment with HER-2 Receptor Inhibitors

It has been proposed that SDC, like breast cancer, can be divided into hormone receptor-positive, ERBB2 (HER-2)-positive, and basal-like groups. Recent reports reveal that the molecular transcriptome landscape of SDC resembles apocrine breast cancer genetic expression profiling, proposing a common signaling and therapeutic approach in these malignancies [17]. Breast cancer and SDC show similar expressions of several biomarkers, such as mammaglobin, GATA3, estrogen receptor (ER), HER-2, PI3KCA, and HRAS. Immunoprofiling of our patient cohort showed expression of mammaglobin and GATA3 in 55% and 100% of the tested patients (five of nine and seven of seven), respectively. Our data also showed the presence of ER-positivity in four female patients (57% of female patients). Multiple studies have shown the prognostic significance of ER-positivity in breast cancer. Furthermore, recent studies have shown an association of ER expression with survival in head and neck cancers [33,34].

The amplification of human epidermal growth factor-2 (ERBB2/HER-2), a major proliferation factor and therapeutic target in breast cancer, has been detected in 20–30% of SDC cases and is associated with poor survival [35,36,37]. Our data show HER-2 expression in 20% of cases, consistent with other studies reporting this association. Further supporting the similarities between the two cancer types is the fact that one of the most common mutations in breast cancer is detected in the PI3KCA gene, which is considered prognostic for targeted anti-PI3K and anti-HER-2 therapy [38]. The activation of the PI3KCA protein requires coupling with other tyrosine kinases, including HER-2, to promote cell proliferation [39]. Thus, clinical assessment of the ERRB2 and/or PI3KCA mutational and expression statuses may serve as prognostic factors and provide a setting in which combination PI3K/ERBB2 inhibition would be effective for SDC.

Trastuzumab (HER-2 monoclonal antibody) has been tested in SDC with studies showing clinical response [40,41,42,43,44,45]. The majority of studies evaluating the benefit of trastuzumab for SDC reserved its use for patients who developed recurrence or metastasis. Accordingly, NCCN Guidelines Version 2.2024 recommend testing for HER-2 receptor status in patients with metastatic SDC but not in the upfront setting when treating curatively [14]. A subset of studies have assessed outcomes for patients treated with trastuzumab concurrently with RT or adjuvantly after completing treatment but before developing metastasis [17,19,40]. The number of patients treated with this approach in these studies is small; however, the results are promising.

While data assessing HER-2-targeted therapy for SDC are limited, its benefits have been proven in prospective fashion in the treatment of breast cancer [46,47,48,49,50]. Ongoing trials aim to de-escalate treatment for low-risk patients by eliminating chemotherapy and treating with HER-2-targeted therapy alone. Other trials have studied the benefits of treatment escalation using ado trastuzumab emtansine (T-DM1), tucatinib, and dual HER-2 blockade with trastuzumab plus pertuzumab [51,52,53,54,55,56,57,58]. Additional investigation evaluating the benefit of HER-2-targeted therapy in the upfront setting and treatment escalation/de-escalation using these alternative agents is warranted to assess the benefits of these approaches for SDC.

### 4.3. Limitations and Future Direction

A limitation of our data is the small cohort size. This was primarily due to the select inclusion criteria of patients with AR+ SDC of the parotid, which represents a rare entity. However, this also allowed for a more homogenous group of patients than other studies reporting outcomes for SDC. While other series did report outcomes for larger cohorts, many included AR-negative patients, limiting the potential conclusions one can draw for outcomes specific to SDC. Although small sample size for this rare entity precludes formal statistical assessment of association between immunoprofiles and outcomes, the vignettes discussed above highlight the importance of molecular subtypes for prognostication and treatment selection.

Based on the current findings of our study, our team initiated prospectively collected tumor tissue banking to perform further detailed molecular profiling on these malignancies. Over time, our goal is to utilize this data to better provide an individualized treatment strategy for these patients and continue to improve response to therapy. We plan to expand upon the cohort analyzed within this manuscript in the future to further assess association with individual molecular profiles on control and survival, and more importantly, on response to various non-chemotherapy systemic agents. We hope that this may allow for personalization of treatment in the future, using molecular profiling to determine selection and sequencing of each therapeutic intervention.

## 5. Conclusions

SDC is a subgroup of rare high-grade salivary gland tumors that portend a poor prognosis, with distant metastasis representing the most common form of failure. We conducted immunoprofiling on patients with AR+ SDC, analyzing 24 separate molecular markers. We found that immunoprofiles, such as expression of HER-2, were highly variable, highlighting the molecular variability of this rare entity and potential importance of treatment individualization. Studies with larger patient numbers using tumor-specific molecular profiling and tumor heterogeneity analyses are justified to better understand the biology of these tumors and define the optimal treatment approach, and ultimately to answer whether the addition of biological agents in the definitive setting against specific molecular targets such as AR and HER-2/Neu will further improve outcomes for these patients.

## Figures and Tables

**Figure 1 cancers-16-01204-f001:**
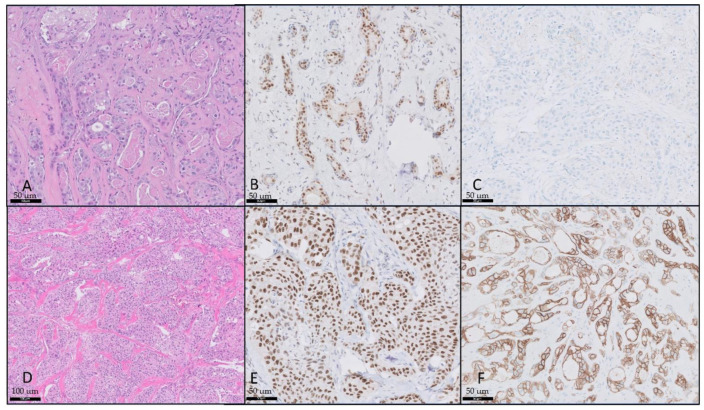
Salivary duct carcinoma with HER-2 and androgen receptor. (**A**) Salivary duct carcinoma, hematoxylin and eosin (H&E) stain, 20×; (**B**) androgen receptor, 20×; (**C**) HER-2 negative (0+), 20×; (**D**) salivary duct carcinoma, H&E, 10×; (**E**) androgen receptor, 20×; (**F**) HER-2 positive (3+), 20×.

**Figure 2 cancers-16-01204-f002:**
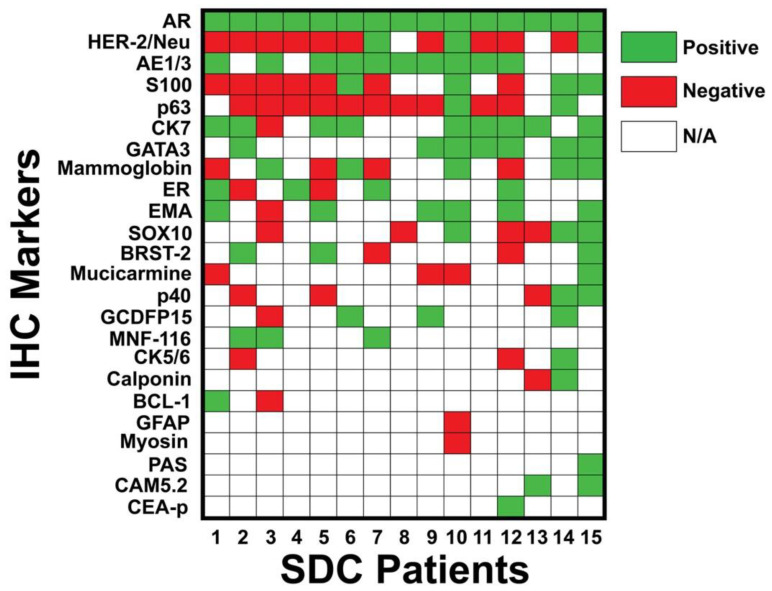
Immunoprofile of each patient. Molecular markers were positive in the following percentage of patients: AR (100%), HER-2/Neu (20%), AE1/3 (67%), S100 (27%), p63 (13%), CK7 (60%), GATA3 (47%), mammaglobin (33%), ER (27%), EMA (40%), SOX10 (20%), BRST-2 (20%), mucicarmine (7%), p40 (13%), GCDFP15 (20%), MNF-116 (20%), CK5/6 (7%), calponin (7%), BCL-1 (7%), GFAP (0%), myosin (0%), PAS (7%), CAM5.2 (13%), and CEA-p (7%). Green = positive; red = negative; white = not tested. Abbreviations: IHC = immunohistochemistry; SDC = salivary duct carcinoma.

**Figure 3 cancers-16-01204-f003:**
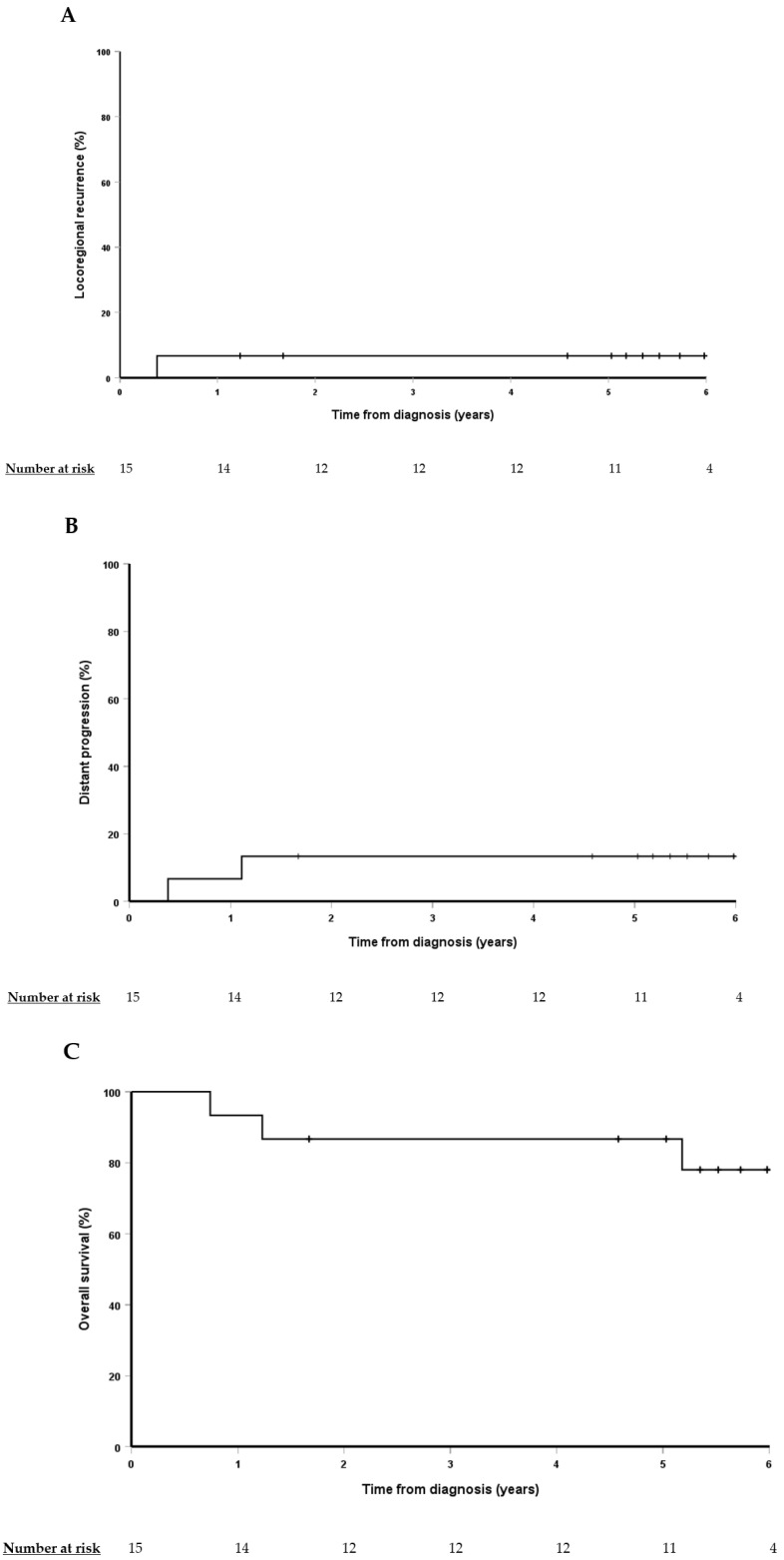
Kaplan–Meier curves for locoregional recurrence (**A**), distant progression (**B**), and overall survival (**C**).

**Table 1 cancers-16-01204-t001:** Demographics and treatment characteristics (*n* = 15).

Characteristics	N (%)
Age: median (IQR)	62 (57–77)
Sex	
Male	8 (53)
Female	7 (47)
Smoking history > 10 pack-years	
Yes	6 (40)
No	9 (60)
Primary site	
Right parotid	8 (53)
Left parotid	7 (47)
Tumor stage	
T1	3 (20)
T2	2 (13)
T3	1 (7)
T4a	6 (40)
T4b	3 (20)
Nodal stage	
N0	7 (47)
N1	1 (7)
N2a	0
N2b	7 (47)
Treatment modality	
Surgery and adjuvant RT + systemic therapy	10 (67)
Surgery and adjuvant RT alone	4 (27)
Definitive radiation + systemic therapy	1 (7)
Surgical margin status	
Negative margins	6 (43 *)
<1 mm margin	4 (29 *)
≥1 mm margin	2 (14 *)
Positive margins	8 (57 *)
No surgery	1 (7)
Perineural invasion	7 (50 *)
Lymphovascular invasion	7 (50 *)
Lymph node involvement: median (IQR)	
Number positive	6 (4–9) ^†^
Number dissected	31 (20–36)
% positive/dissected	18 (7–25) ^†^
Extranodal extension	5 (71) ^†^
Radiation target	
Primary site + ipsilateral neck	12 (80)
Primary site alone	3 (20)
Radiation prescription dose	
6000 cGy	3 (20)
6600 cGy	10 (67)
6996 cGy	2 (13) ^‡^
Systemic therapy	
None	4 (27)
Cisplatin	3 (20)
Carboplatin + Paclitaxel	3 (20)
Carboplatin	2 (13)
Gonadotropin releasing hormone analog	2 (13)
Non-steroidal androgen receptor inhibitor	1 (7)

Abbreviations: IQR = interquartile range; RT = radiation therapy. * Percentage of patients who underwent surgery (*n* = 14) ^†^ Numbers listed pertain only to patients with positive lymph nodes on surgical pathology (*n* = 7) ^‡^ One patient was treated with definitive radiation, and one underwent resection with gross residual noted postoperatively.

**Table 2 cancers-16-01204-t002:** Radiation treatment toxicities.

Toxicities	Grade 0 *n* (%)	Grade 1 *n* (%)	Grade 2 *n* (%)	Grade 3 *n* (%)
Acute				
Fatigue	0	9 (60)	3 (20)	3 (20)
Pain	1 (7)	3 (20)	10 (67)	1 (7)
Dysphagia	5 (33)	9 (60)	1 (7)	0
Odynophagia	3 (20)	7 (47)	3 (20)	2 (13)
Dysgeusia	2 (13)	2 (13)	11 (73)	0
Xerostomia	0	7 (47)	3 (20)	5 (33)
Thick secretions	0	13 (87)	1 (7)	1 (7)
Radiation dermatitis	0	6 (40)	9 (60)	0
Mucositis	2 (13)	2 (13)	9 (60)	2 (13)
Late				
Fatigue	11 (73)	3 (20)	1 (7)	0
Pain	11 (73)	1 (7)	1 (7)	2 (13)
Dysphagia	12 (80)	2 (13)	0	1 (7)
Odynophagia	15 (100)	0	0	0
Dysgeusia	10 (67)	4 (27)	1 (7)	0
Xerostomia	6 (40)	8 (53)	1 (7)	0
Thick secretions	7 (47)	7 (47)	1 (7)	0
Radiation dermatitis	9 (60)	6 (40)	0	0
Mucositis	15 (100)	0	0	0

**Table 3 cancers-16-01204-t003:** Summary of studies reporting outcomes for locally advanced salivary duct carcinoma.

Author (Year Publication) Study Period Institution (Country)	Patients (*n*)	Tumor Site (%)	Markers tested	AR+ (%)	HER-2/Neu+ (%)	Surgery (%)	Received RT RT Technique Nodal Irradiation (%)	Median RT Dose (Gy)	Non-ADT Systemic Therapy (%) Systemic Agent (*n*)	ADT (%) ADT Agent (*n*)	Median F/U (mo)	LR (%)	RR (%)	DR (%)	OS * (%)
Jaspers et al. (2011) [15] 2000–2010 Radboud University Nijmegen Medical Centre (Netherlands)	10	Parotid: 80 SMG: 20	AR HER2	100	NR	40	40 NR NR	NR	20 (after failure to ADT) Docetaxel (2)	100 (first-line therapy after progression) Bicalutamide (10) GnRH analog (1)	NR	10	80	90	30
Masubuchi et al. (2014) [16] 2005–2012 International University of Health and Welfare Mita Hospital (Japan)	32 (3 had metastatic disease)	Parotid: 78 SMG: 13 Sublingual: 3 Oral cavity: 6	AR HER2 EGFR	75	44	94	37 NR NR	NR	25 NR	0	22	NR	NR	NR	2-yr: 73
Luk et al. (2016) [8] 1989–2014 Royal Prince Alfred Hospital (Australia)	23	Parotid: 78 SMG: 17 Sublingual: 4	19 markers	70	30	100	96 NR NR	NR	39 Cisplatin (6) Carboplatin (1)	0	26	NR	NR	43	5-yr DFS: 36 5-yr DSS: 43
Dalin et al. (2016) [17] 2000–2015 MSKCC (USA)	16	Parotid: 94 SMG: 6	410 markers	75	35	94	94 NR NR	NR	31 Trastuzumab (2 concurrent w/RT, 1 adjuvant, 2 after progression)	25 (after progression) Bicalutamide and/or GnRH analog (4)	NR	19	NR	44	38
Haderlein et al. (2017) [18] 1998–2016 University Hospital of Erlangen (Germany)	67 (45 pts with F/U data)	Parotid: 88 SMG: 9 Minor salivary: 3	AR HER2 ER PR EGFR PD-L1	84	25	67	57 NR Yes, omitted only if ≤pT2N0	64	36 Chemotherapy (specific agent NR)	0	26	11	13	33	3-yr: 72 5-yr: 57
Adeberg et al. (2019) [19] 2010–2017 Heidelberg Ion-Beam Therapy Center (Germany)	28	Parotid: 79 SMG: 7 Minor salivary: 7 Sublingual: 4 Lacrimal: 4	AR HER2	68	50	82	100 IMRT + Carbon ion boost 100 (levels II-III only)	IMRT: 54 Carbon ion boost: 18	18 (adjuvant) Trastuzumab (5)	43 (adjuvant) Bicalutamide (12)	30	14	11	32	Median 93 months
Laughlin et al. (2023) [13] 1971–2018 Mayo Clinic (USA)	89	Parotid: 87 SMG: 10 NR: 3	AR HER2	88	61	96	87 2D (1), 3D (42), IMRT (55), proton (3) Yes, for N+	61	29 Cisplatin (19)	0	44	27	NR	44	42
Present Series 2013–2019	15	Parotid: 100	24 markers	100	20	93	100 IMRT 80	66	53 Cisplatin (3) Carboplatin/paclitaxel (3) Carboplatin (2)	20 GnRH analog concurrent w/RT (2) Bicalutamide after progression (1)	66	6	0	13	5-yr: 87

Abbreviations: AR+ = androgen receptor positive; RT = radiation therapy; ADT = androgen deprivation therapy; F/U = follow-up; LR = local recurrence; RR = regional recurrence; DR = distant recurrence; OS = overall survival; SMG = submandibular gland; NR = not recorded; GnRH = gonadotropin releasing hormone; EGFR = epidermal growth factor receptor; DFS = disease-free survival; DSS = disease-specific survival; MSKCC = Memorial Sloan Kettering Cancer Center; ER = estrogen receptor; PR = progesterone receptor; PD-L1 = programmed death-ligand 1; IMRT = intensity-modulated radiation therapy. * Outcomes are reported as crude rates unless otherwise specified.

## Data Availability

Research data are stored in an institutional repository and will be shared upon request to the corresponding author.

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
