# Peer review of "Immunoprofiles and Oncologic Outcomes of 15 Patients with Androgen Receptor-Positive Salivary Duct Carcinoma"

_cancers, 2024, doi:10.3390/cancers16061204_

Round 1

Reviewer 1 Report

Comments and Suggestions for Authors

The authors investigated the immunoprofiles of salivary duct carcinoma to esbtablish the correlation of Androgen Receptor expression with the diagnosis of salivary gland carcinoma. Such work would benefit the understanding on the biological features of tumors. However, there are a lot of defects in the manuscript that have to be corrected to satisify the publication quality.

1. the subject have to be revised, because it sounds like a topic of review article.

2. Please re-organize the data as a research article in order to outline the significane of AR or HER2 in salivary gland carcinoma.

3. To ensure the conclusion, usually one or more statistical approaches were supposed to be applied. 

Comments on the Quality of English Language

The language requires minor progression. Moreover, there are lots of semicolons in the text, where are supposed to be replaced with comma.

Reviewer 2 Report

Comments and Suggestions for Authors

Summary Statistics for Patient Characteristics: The manuscript currently presents summary statistics for patient characteristics as median and ranges. Given the small size of the cohort, it is advisable to use median and interquartile range (IQR) instead of median and range. Median and IQR provide a better understanding of the data distribution and variability within a small cohort.

Table 1 Presentation: The layout of Table 1 makes it difficult to read, with names and figures centered rather than aligned to the left. For improved readability and to facilitate a quicker understanding of the data, consider aligning all text to the left.

Figure 2 Clarification Needed: The role and contribution of Figure 2 to the overall findings are unclear. A brief explanation or clarification within the text on how Figure 2 complements the study's results would enhance the reader's comprehension.

Figure 3A and B - Survival Curves: The survival curves presented in Figure 3A and B seem to be based on too few events.

Table 3 Legibility: Table 3 is currently difficult to read. Ensuring that all tables are clearly legible and formatted for easy interpretation is crucial. This might involve adjusting the font size, spacing, or layout.

Study Cohort and Novelty of Data: The study involves a very small number of patients and offers limited new data. Enhancing the literature review section could add depth to the article by providing a more comprehensive context for the study's findings, especially regarding toxicity results. A thorough literature review could also establish a clearer connection with the obtained results, highlighting the study's contribution to the field despite its limitations due to the small cohort size.

By addressing these points, the manuscript could significantly improve in clarity, readability, and scholarly contribution.

Reviewer 3 Report

Comments and Suggestions for Authors

The manuscript titled ' Androgen Receptor Positive Salivary Duct Carcinoma: Immunoprofiling, Oncologic Outcomes, and Associated Literature Review’ assess and report clinical outcomes, patterns of failure to treatment, and immunoprofiles for patients diagnosed with AR+ SDC of the parotid gland treated with curative intent. The manuscript requires a few minor improvements to enhance clarity and impact, making it suitable for publication.

1. Please simply the title of manuscript.

2. Please provide scale bar to all photographic/IHC images.

3. Please provide better resolution image for fig.3 as its difficult to interpret.

4. The manuscript requires a significant attention to improve punctuations, grammar and the readability.

Comments on the Quality of English Language

Please see comment 4 above. 

Round 2

Reviewer 1 Report

Comments and Suggestions for Authors

All conerns have been addressed. I have no more question.

Reviewer 3 Report

Comments and Suggestions for Authors

In the updated manuscript Androgen Receptor Positive Salivary Duct Carcinoma: Immunoprofiling, Oncologic Outcomes, and Associated Literature Review”, the authors did address all the previous concerns and now the manuscript is convincing and would help advance the understanding of role of Androgen Receptor in Cancers. The manuscript is now updated, and I recommend this article for publication.